# Rapid Progression of COVID-19-Associated Fatal Capillary Leak Syndrome

Eva Novotná [1,*], Pavlína Filipová [1], Ivan Vonke [2], Bohuslav Kuta [3] and Aleš Chrdle [1,4,5]

1    Department of Infectious Diseases, České Budějovice Hospital, 37001 České Budějovice, Czech Republic
2    Department of Clinical Haematology, České Budějovice Hospital, 37001 České Budějovice, Czech Republic
3    Department of Cardiac Surgery, České Budějovice Hospital, 37001 České Budějovice, Czech Republic
4    Tropical and Infectious Diseases Unit, Royal Liverpool University Hospital, Liverpool L7 8YE, UK
5    Faculty of Health and Social Sciences, University of South Bohemia in České Budějovice, 37001 České Budějovice, Czech Republic
*    Correspondence: novotna.eva@nemcb.cz; Tel.: +420-(38)-7874601

**Abstract:** Several cases of capillary leak syndrome (CLS) related to COVID-19 or vaccination against SARS-CoV-2 have been described in the literature. We present a case of a 42-year-old, previously healthy male, presenting with a mild form of COVID-19, who suddenly developed severe shock with hypotension and severe hemoconcentration within hours of admission to the hospital. Volume resuscitation was not effective, increasing hemoglobin (198 g/L on admission, 222 g/L 9 h later) suggested fluid leak into peripheral tissues. After cardiac arrest, the patient was resuscitated and connected to extracorporeal membrane oxygenation, but died shortly afterwards due to refractory heart failure. Retrospective investigation of blood samples confirmed diagnosis of CLS by progressive hypoalbuminemia (40 g/L on admission, 14 g/L 19 h later) and monoclonal gammopathy kappa (4.7 g/L). Patient's CLS was triggered by COVID-19, either a first attack of idiopathic CLS called Clarkson's disease or a COVID-19-induced secondary CLS.

**Keywords:** capillary leak syndrome; COVID-19; refractory shock; hemoconcentration

## 1. Introduction

Capillary leak syndrome (CLS) is a rare disease with a potentially fatal outcome. The pathological mechanism involves the leakage of fluid and protein from the bloodstream into peripheral tissues, resulting in hypovolemia, hypoalbuminemia, elevated hematocrit with hemoconcentration, oedemas and hypotension. The diagnosis is based on the presence of the triad: hypotension, hemoconcentration and hypoalbuminemia [1]. There have been cases of capillary leak syndrome reported in both COVID-19 as well as vector-based vaccination against SARS-CoV-2 [2–7]. We present a case of a severe, rapidly progressing manifestation of capillary leak syndrome associated with a mild form of COVID-19.

## 2. Case Report

The patient was a previously fit and well 42-year-old white male, with no significant past medical history. He was admitted to the hospital in February 2022 from home because of a syncope and hypotension. He complained of joint pain lasting several hours, fatigue and non-productive cough lasting for two days. He had tested positive for COVID-19 two days prior (he had no history of prior vaccination against SARS-CoV-2).

On admission, the patient had hypotension 90/60 mmHg, heart rate was regular, 65 bpm, GCS was 15/15, there were no obvious swellings and he appeared well perfused. Pulse oxymetry showed O2 saturation 96% on room air. Blood count showed an increase in hemoglobin level (198 g/L) and hematocrit (0.57). Other routine blood tests were unremarkable and abnormalities at the time of admission are shown in Table 1.

**Table 1.** Laboratory values during admission in a patient with rapid progression of COVID-19-associated capillary leak syndrome.

| Test | Reference Range | Time of Admission | ICU: 9 h Post Admission | ICU: 19 h Post Admission |
|---|---|---|---|---|
| Hemoglobin | 135–175 g/L | 198 | 222.0 | 200 |
| Hematocrit | 0.40–0.50 | 0.57 | 0.64 | 0.62 |
| Erythrocytes | $4.0$–$5.8 \times 10^{12}$/L | 6.18 | 7.01 | 6.33 |
| Leukocytes | $4.0$–$10.0 \times 10^{9}$/L | 8.40 | 12.80 | 31.7 |
| Platelets | $150$–$400 \times 10^{9}$/L | 239 | 222 | 169 |
| Albumin * | 35–53 g/L | 40.0 | NA | 14.0 |
| Creatinine | 64–104 μmol/L | 108 | 111 | 180 |
| Alanine transaminase | 0.10–0.78 μkat/L | 0.60 | 0.41 | 7.54 |
| Creatine kinase | 0.05–0.72 μkat/L | 2.38 | 2.37 | 12.68 |
| C-reactive protein | <2 mg/L | 8.6 | 4.1 | <2 |
| Lactate | 0.6–2.3 mmol/l | NA | 4.4 | 16.6 |
| pH | 7.36–7.44 | NA | 7.36 | 7.29 |
| pO2 | 9.9–14.4 kPa | NA | 11.4 | 17.2 |
| pCO2 | 4.8–5.9 kPa | NA | 3.9 | 4.4 |
| Paraprotein IgG type kappa * | 0 g/L | 4.6 | | |
| SARS-CoV-2 IgG | <12.9 AU/mL | 2.2 | | |
| SARS-CoV-2 N-antigen | <8.92 pg/mL | <8.92 | | |

NA–not available, ICU–intensive care unit.* Retrospectively tested serum samples were considered stable as they were stored at 2–8 °C for less than 7 days according to the standard operating procedure.

His ECG was unremarkable with normal sinus rhythm 72 bpm. After arrival at the hospital, he shortly fainted two more times. Extended initial imaging work-up (chest X-ray, point of care abdominal ultrasound and transthoracic echocardiography) did not find any pathology except for ventricular septal wall thickness of 12 mm on transthoracic echocardiography. As the patient was COVID-19-positive and symptomatic, he was admitted to the COVID-19 dedicated unit. Since the patient reported reduced oral fluid intake during the last few days (estimated less than 1 L per day), dehydration was initially considered to be the main cause of hypotension and hemoconcentration.

The patient was given 4500 mL of intravenous eletrolyte solutions over the next 9 h, while he remained fully alert with no more fainting and no other disturbing symptoms. His blood pressure, however, was persistently low at 90/60 mmHg with heart rate 80 bpm, and his urine output was reduced (around 50 mL/hour). Follow-up blood count after 9 h in the hospital showed progressive elevation of hemoglobin (222 g/L) and hematocrit (0.64). The rate of hematocrit and hemoglobin increase were concerning in the light of recent high volume fluid replacement. He was transferred to the intensive care unit. At this point, albumin levels were retrospectively tested from blood drawn on admission to the hospital. The total protein level was decreased (57.0 g/L) and the albumin level was normal (40.0 g/L).

Up to this time the patient was fully alert and did not appear unwell. Shortly after transfer to the ICU he became plethoric and started to complain of leg pain and chest pain. Blood pressure remained around 100/60 mmHg, heart rate was 90–100 bpm, GCS was still 15/15, saturated 96% on room air and urine output was 50 mL/hour without vasopressor support or administration of diuretics. His leg pain worsened considerably over the following two hours, and therefore we considered COVID-19-related acute arterial occlusion in the lower extremities; a CT angiography of the aorta and lower limbs together with CT angiography of the pulmonary artery was performed. All examined arteries of the systemic and pulmonary circulation were fully patent. We continued to administer intravenous electrolyte solutions, and vasopressors were discussed but withheld at this time.

Eighteen hours after admission to the hospital, the patient became suddenly unresponsive, GCS dropped from 15/15 to 3/15, blood pressure remained 100/60 mmHg, heart rate increased to 120 bpm, urine output dropped below 30 mL/h. The patient was intubated

and ventilated, vasopressor support with noradrenaline was started. Hemoglobin and hematocrit levels remained high (Table 1). CT of the brain was performed, and no brain pathology was found.

Within less than half an hour, the patient suffered cardiac arrest (pulseless electric activity) and chest compressions were started. The patient was placed on arterio-venous extracorporeal membrane oxygenation (A-V ECMO), however, after brief stabilization, circulatory collapse reoccurred and the patient died. On autopsy, no apparent pathology was detected with the exception of a mild cardiomyocyte hypertrophy as a possible sign of chronic heart failure.

Retrospectively, the next day, IgG kappa monoclonal gammopathy was detected in serum (4.6 g/L), and albumin level in blood samples from the period of sudden deterioration while in ICU was reduced to 14 g/L.

## 3. Discussion

We report an extremely rapid course of fatal capillary leak syndrome related to a mild form of COVID-19, as clinical presentation before admission was unremarkable and did not suggest a more severe form of COVID 19. There are two forms of CLS, an idiopathic form (ICLS–Clarkson's disease) and secondary CLS. Secondary CLS can be induced by drugs, infection (often viral), malignancies or autoimmune disease. Several cases of CLS triggered by COVID-19 have been reported [2–4]. Cases of CLS related to vaccination against SARS-CoV-2 were reported, mainly in vector-based vaccines and represent contraindication of those vaccines [5–7].

Although the pathophysiology is not completely clear, a transient severe endothelial disruption caused by multiple factors plays a major role [1].

Monoclonal immunoglobulin (mostly IgG1 with kappa light chains) in serum has been reported in 70–95% of adult cases of the idiopathic form. The role of this factor in the pathophysiology of CLS is unknown [8].

The mortality rate of acute CLS is 20–30%. The cause of death may be cardiac arrest during hypovolemic shock, multiorgan failure due to hypoperfusion, or pulmonary embolism due to hemoconcentration. In the literature, progression within a few days has been reported [3,9].

Treatment is empirical and experience-based. As fluid overload may lead to increased mortality during the recovery phase, prudent intravascular expansion (crystalloids first, followed by albumin) with permissive hypotension is required to avoid compartment syndrome while ensuring sufficient organ perfusion [1]. Aminophylline/teophylline and terbutaline have been successfully used for idiopathic CLS flares [10] and intravenous immunoglobulins (IVIG) administered monthly have been successfully used to prevent ICLS flares [1].

Low serum albumin on admission has been reported as a negative prognostic factor of poor outcome for COVID-19 [11]. There are multiple factors affecting albumin levels apart from plasma leak to the interstitial space, including reduced dietary intake during COVID-19, acute and chronic inflammation, reduced synthesis in the liver or loss via skin, gastrointestinal tract, or kidneys, or in many cases a combination of several of the above as a sign of generally poor health and significant co-morbidities [11]. The serum albumin level in our case taken on admission to the hospital, however, was within the limits of normality but dropped rapidly afterwards. Oedema was not observed until later in the ICU, compatible with aggressive fluid replacement.

On admission, our patient had only hypotension and hemoconcentration, but no edemas and no hypoalbuminemia, which could be more suggestive of CLS. The diagnosis was initially unclear, as the clinical presentation and laboratory results did not suggest sepsis or mere dehydration. Endocrinopathies, namely hypocorticalism, have been reported to cause persistent hypotension in COVID-19 cases [12]. Interestingly, his serum electrolyte levels remained within the limits of normality throughout his hospital stay. As hemoglobin level was high despite aggressive fluid replacement, the main differential diagnosis became

the polycythemia range of diseases. However, the rapid increase in hemoglobin and hematocrit levels was not fully compatible with a chronic condition.

The patient died from refractory cardiac arrest. Only after reviewing the symptoms retrospectively did we conclude the diagnosis of CLS, which was confirmed by retrospectively requested albumin level from the sample taken after the patient's condition deteriorated in ICU.

Several cases of CLS related to COVID-19 have been described in the literature, both secondary CLS and manifestations of Clarkson's disease. Similar to our case, all reported cases had a mild to moderate course of COVID-19 [2–4]. We believe that COVID-19 was a contributing factor to the patient's CLS. The patient may have had either a first attack of idiopathic CLS, which would be supported by the presence of monoclonal gammopathy, or secondary CLS induced by COVID-19 with incidental monoclonal gammopathy.

This case report confirms previous reports that CLS as a rare disease is often diagnosed late [13] or misdiagnosed [9,13,14].

Previously unrecognized chronic heart failure detected at autopsy may have contributed to the rapid progression and fatal outcome, although echocardiography on admission did not suggest cardiac dysfunction.

Another factor contributing to the delayed diagnosis in our case was normal albumin level in the initial sample. Normal albumin levels on admission to hospital have been described in one case of fatal CLS syndrome triggered by COVID-19 [3].

We conclude that this and similar cases put in perspective the occurrence of CLS which is not only an adverse effect of vaccination but also a complication of mild COVID-19. Therefore, it is not the vaccination itself but rather an erratic immune system response to S-antigen of SARS-CoV-2 both during natural infection and post-vaccination that induces CLS.

To improve the diagnosis of CLS associated with COVID-19 (or associated with vaccination against SARS-CoV-2) we propose testing for monoclonal gammopathy in clinical scenarios compatible with CLS concurring with COVID-19.

**Funding:** This research received no external funding.

**Institutional Review Board Statement:** Not applicable.

**Informed Consent Statement:** Patient consent was waived due to patient´s death.

**Data Availability Statement:** Not applicable.

**Conflicts of Interest:** The authors declare no conflict of interest.

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
