# Peer review of "Rapid Progression of COVID-19-Associated Fatal Capillary Leak Syndrome"

_2036-7449, doi:10.3390/idr14060088_

Round 1

Reviewer 2 Report

This manuscript presents an interesting case report but is limited in the description of the case - it does not classify the cough as productive or not, and it is not stated in the anamnesis if the patient has already had CLS (only in the abstract). Besides, the general examination of the patient is incomplete (and incomplete vital signs description). 

  • The report can be better organized concerning patient evolution versus vital signs versus laboratory tests versus treatment;
  • Under what blood storage conditions (temperature) were albumin and IgG kappa measured?
  • Missing references in the introduction;
  • References are missing in paragraphs 94-95 and 96-97;
  • Line 127: confused. Did the authors mean that "... secondary CLS AND MONOCLONAL gammopathy COMBINATION is often diagnosed late..."? It would not make sense. It would make sense to say that CLS is diagnosed late or misdiagnosed;
  • The authors should consider the importance of searching for monoclonal gammopathy in cases of secondary CLS triggered by COVID-19;
  • Reading can be more fluid.

Round 2

Reviewer 1 Report

Line 37: Please fix "against COVID-19" with "against SARS-CoV-2" and check the whole manuscript for this reason.

Lines 120-121: Please change COVID-19 with SARS-CoV-2

Are you sure that this could be defined as "mild COVID-19"?

Table 1: Please write pH in this form (not PH)

Line 175: Check for punctuation

Line 225: Please speak about "vaccination against SARS-CoV-2" not COVID-19

I suggest reading and take in consideration the following articles about COVID-19 complication: 10.3892/br.2022.1517, 10.3892/wasj.2022.149.

Kind regards

Author Response

Thanks for re-reviewing, very helpful.

Line 37: Please fix "against COVID-19" with "against SARS-CoV-2" and check the whole manuscript for this reason. changed here and on few other places accordingly

Lines 120-121: Please change COVID-19 with SARS-CoV-2 - changed here and on few other places accordingly

Are you sure that this could be defined as "mild COVID-19"? explained in discussion

Table 1: Please write pH in this form (not PH) corrected

Line 175: Check for punctuation rephrased

Line 225: Please speak about "vaccination against SARS-CoV-2" not COVID-19 done

I suggest reading and take in consideration the following articles about COVID-19 complication: 10.3892/br.2022.1517, 10.3892/wasj.2022.149. read the articles, but at the this stage were not able to find significantly new information that would warrant incorporating into our manuscript